# Government spending shocks and default risk in emerging markets

**Ming Jiang[1], Jingchao Li[2]***

**1** Antai College of Economics and Management, Shanghai Jiao Tong University, Shanghai, China, **2** School of Business, East China University of Science and Technology, Shanghai, China

* jingchaoli@ecust.edu.cn

## Abstract

The coronavirus pandemic has revived interest in the effects of fiscal policy. This paper studies the effects of government spending on default risk in emerging economies. We first build a general equilibrium small open economy model where government spending shocks influence external debt and sovereign bond spreads. We show that external debt piles up and sovereign bond spreads increase following a government spending shock. We then develop VAR evidence based on a panel of 18 countries. We find that in response to a 10% government spending increase, (1) the real effective exchange rate appreciates by 1.0% and the current account to GDP ratio deteriorates by 0.0025 on impact; (2) external debt increases by an average of 3.5% in the year following the shock; and (3) the EMBI Global spread rises by an average of 25 basis points within two years and peaks at 132 basis points 14 quarters after the shock, suggesting a higher sovereign default risk. The empirical results confirm the theoretical predictions from the general equilibrium model.

## Introduction

In order to combat the coronavirus pandemic, governments all over the world have implemented fiscal actions, including additional spending, tax cuts, and loans and capital injections by the public sector [1]. On the one hand, fiscal support in response to the pandemic has prevented more severe economic recessions. On the other hand, such support has led to larger government deficits and debt. In 2020, average overall deficit as a share of GDP is 11.7 percent for advanced economies, 9.8 percent for emerging market economies, and 5.5 percent for low-income developing countries [1]. Moreover, average public debt worldwide has reached an unprecedented 97 percent of GDP [1]. Countries such as Argentina, Ecuador, Ethiopia, and Lebanon defaulted on their debts. This paper studies the effects of government spending on default risk in emerging economies.

We first build a dynamic stochastic general equilibrium small open economy model with government spending shocks. The quantitative model of this paper is based on Aguiar and Gopinath [2] and Arellano [3]. Distinct mechanisms in these two papers are employed in order to match default frequency and business cycle moments observed in the data: the asymmetric output penalty in case of default is crucial in Arellano [3], whereas trend shocks to output are the key to explaining the default frequency in Aguiar and Gopinath [2]. This paper

**Data Availability Statement:** All relevant data are within the paper and its Supporting information files.

**Funding:** Jiang thanks the National Natural Science Foundation of China (Grant 72103135 and Grant 72231003) for research support. Li thanks the

National Social Science Fund of China (Grant 21AZD036) and the National Natural Science Foundation of China (Grant 72274062) for research support.The funders had no role in study design, data collection and analysis, decision to publish, or preparation of the manuscript.

**Competing interests:** The authors have declared that no competing interests exist.

extends these models to include government spending shocks and focuses on the effects of spending shocks on sovereign default risk.

The sovereign government borrows using one-period, non-contingent discount bonds in the international credit market. Since the government cannot commit to repay its debt, it has to pay an interest rate that is higher than the risk-free rate in order to borrow from foreign lenders. The difference between the interest rate the sovereign pays and the risk-free rate is defined as the sovereign spread. The model is calibrated to Argentina's data and simulated for 20 quarters. We find that following a positive government spending shock, the sovereign spread increases on impact. External debt increases by 0.57% on impact and stays positive for 5 periods after the shock before reverting back to its steady state. The ratio of current account to GDP falls on impact and then turns positive. This simulation exercise shows that a government spending shock raises the default risk in a developing country which borrows from foreign lenders and bears a sovereign spread.

We then carry out empirical analysis and identify government spending shocks using structural vector autoregression (SVAR). There are two main strategies of identifying government spending shocks with aggregate data: SVAR analysis as in Blanchard and Perotti [4] and the narrative approach developed by Ramey and Shapiro [5]. Blanchard and Perotti [4] identify fiscal shocks by assuming that government spending is predetermined within one quarter so that it does not respond to other structural shocks contemporaneously. Ramey and Shapiro [5] add a military build-up dummy variable in the autoregressive model to isolate an exogenous component of government spending. The two methods usually produce different responses to government spending shocks. Specifically, the first approach finds that consumption and real wages rise after a positive government spending shock, consistent with the predictions of New Keynesian models. In contrast, the second approach generates a fall in the two variables which is often consistent with neoclassical models. In this paper, we use a panel VAR to identify government spending shocks. The reason for using a panel VAR is twofold: first, the narrative approach cannot be applied to the panel of developing countries studied in this paper, since it requires that wars be extraterritorial, which is not the case for most of the developing countries; second, the inception of the EMBI Global spread data varies from 1993Q4 to 2009Q4 in different countries, making individual VAR analysis impractical in many countries.

The VAR approach has been criticized for ignoring the "anticipation effect": the VAR-identified fiscal shocks may have been expected by the private sector, who will take action even before the spending is realized. Ilzetzki, Mendoza, and Végh [6] show in their online appendix that this is not the case in developing countries. They argue that agents in developing countries could not have a good estimate of fiscal shocks because fiscal shocks are sufficiently volatile in developing countries. The authors compare the central bank's estimation errors of government consumption with the VAR residuals in four developing countries. Central bank errors are computed as the difference between final statistics on government consumption and the central bank's estimate in the quarter following the time of the government expenditure. The authors find a strong correlation between central bank errors and VAR residuals.

There have also been disputes related to the responses of the exchange rate and the trade balance under the VAR strategy. Using Cholesky decomposition, Kim and Roubini [7] conclude that the exchange rate depreciates and that the trade balance improves after a fiscal shock. With a different identification scheme, Monacelli and Perotti [8] find that the trade balance deteriorates, although the exchange rate still depreciates. Ilzetzki, Mendoza, and Végh [6] examine the effects of fiscal stimulus for different groups of countries and episodes, and suggest that different country characteristics, such as the exchange rate regime and openness to trade, lead to quite different responses of macroeconomic variables. However, none of the empirical analysis takes into consideration the risk of sovereign default, which is an innovative

and important feature of this paper. Since a higher level of government spending motivates the sovereign state to borrow more from abroad, it tends to induce a higher level of sovereign debt, which in turn may affect the country's default decisions. Therefore, it is important to study the effects of government spending shocks in an environment where sovereign states can default.

In our empirical analysis, we find that following a government spending shock, the EMBI Global spread rises significantly. GDP increases by 1.33% on impact. However, the responses of GDP are negative in the long run, implying that the increase in government spending is more-than-fully crowded out by the decrease in the other components of GDP. This result implies that higher external debt and default risk may prevent fiscal policies in developing countries to be effective. We then divide the sample into episodes of fixed exchange rate regime and episodes of flexible exchange rate regime. We find that the increase of default risk in countries with fixed exchange rate tends to be larger compared to those with flexible exchange rate regimes. We also differentiate between Latin American countries and European countries and find that the increase of default risk in Latin American countries is much larger.

This paper has two main contributions. First, models of sovereign default, based on the seminal framework in Eaton and Gersovitz [9] and later extended by Aguiar and Gopinath [2], Arellano [3], and Arellano, Mateos-Planas, and Ríos-Rull [10], typically abstract from fiscal policy. We contribute to this literature by adding government spending shocks to a sovereign default model. In this sense, our paper is most related to Espino et al. [11], who investigate optimal fiscal and monetary policies for emerging markets in a sovereign default model. Our paper differs from Espino et al. [11] notably by building an econometrics model and carrying out empirical analysis.

Second, researchers have showed that government spending is usually less effective in stimulating output in developing countries than in advanced countries [6, 12]. Some obstacles preventing fiscal policy to be effective in developing countries include the procyclicality of fiscal policy [13], foreign aid financing of government spending [14, 15], and low marginal investment efficiency [16]. We contribute to this literature by showing that increased default risk is also a factor that could weaken the stimulus effect of government spending in the long run.

## Literature review

This paper links two distinct strands of research. The first is applied theory on sovereign default with technology shocks. A standard model with endogenous default originates from the work of Eaton and Gersovitz [9]. The model is then extended by Aguiar and Gopinath [2] who highlight the role of the stochastic trend in emerging markets, Arellano [3] who emphasizes that default is more likely in recessions, and Mendoza and Yue [17] who endogenize the income process and the default cost. More recently, Asonuma and Joo [18] examine multi-round renegotiations between a risk averse sovereign debtor and a risk averse creditor. They show that high creditor income results in both longer delays in renegotiations and smaller haircuts. Arellano, Mateos-Planas, and Ríos-Rull [10] model debt crises with partial default and an endogenous length. The authors find that partial default is an amplifying force for debt crises. Hatchondo, Martinez, and Sosa-Padilla [19] and Arellano, Bai, and Mihalache [20] examine the impact of COVID-19 on default risk in emerging markets. Both papers find that in response to unexpected shocks, the highest social gains are achieved if haircuts are coupled with debt service suspensions in a standard sovereign default model.

Researchers have also studied heterogeneity in defaults. Erce and Mallucci [21] inspect selective defaults that discriminate across investors. They find that domestic defaults are more likely in countries with smaller credit markets whereas external defaults are more likely in

countries with depressed import markets. Wicht [22] shows that while multilateral debt could reduce default risk, it raises the subordination risk of private liabilities. The aforementioned studies have examined the dynamics between sovereign debt and default risk. However, to what extent a government spending expansion will influence a nation's default risk is not well understood. This paper fills the gap by investigating the effects of government spending shocks in an environment where a country cannot commit to repay its debt.

The second line of literature related to this paper is empirical work on the effects of government spending shocks. There are three broad categories of empirical approaches used to estimate government spending multipliers [23]. The first approach is aggregate country-level time series or panel estimation. Examples include Auerbach and Gorodnichenko [24], Blanchard and Perotti [4], Hall [25], Ramey [26], and Ramey and Zubairy [27]. With proper strategy, clean identification of aggregate effects can be achieved. However, relying on only aggregate data sometimes raises issues of weak instruments or else that standard errors on response coefficients are simply too large.

The second approach is estimated or calibrated New Keynesian dynamic stochastic general equilibrium (DSGE) model, such as Cogan et al. [28], Coenen et al. [29], Zubairy [30], Leeper, Traum, and Walker [31], and Sims and Wolff [32]. Counterfactuals can be performed within the DSGE framework. But this approach hinges critically on the details of the general equilibrium model.

The third approach is subnational geographic cross-section or panel estimation. Examples include Nakamura and Steinsson [33], Serrato and Wingender [34], Dupor and Guerrero [35], Chodorow-Reich [36], and Auerbach, Gorodnichenko, and Murphy [37]. More data can be used and there is a greater scope for finding instruments. However, this approach estimate relative rather than aggregate multipliers because of spillovers among different regions. Overall, each approach has its strengths and weaknesses. There is no best approach.

## Economic model

The sovereign debt and default model is based on the seminal framework of Eaton and Gersovitz [9] and quantitative models of sovereign default developed later by other researchers such as Aguiar and Gopinath [2], Arellano [3], and Arellano, Mateos-Planas, and Ríos-Rull [10]. While previous work on sovereign default mainly examines productivity shocks, this paper focuses on the role of government spending shocks in affecting default risk and sovereign spread. As the empirical evidence shows in the next section, a positive government spending shock has non-neglectable influence on sovereign spread. Therefore, it is very important to include government spending shocks in a sovereign default model and to investigate the mechanism through which spending shocks affect default risk.

### The model economy

The economy is inhabited by an infinitely-lived, risk averse representative household who values consumption and leisure. The government maximizes the household's utility and borrows on behalf of it in the international credit market. The government can choose to default depending on the debt level of the sovereign ($d$), its productivity ($z$), and government spending ($g$). Let $V_{pay}(d, z, g)$ denote the value of repaying its debt, and $V_{def}(z, g)$ the value of default. In each period, given the initial debt level and observing the productivity shock and the spending shock, the government chooses the larger value between repaying its debt and default:

$$V(d, z, g) = \max\{V_{pay}(d, z, g), V_{def}(z, g)\}. \tag{1}$$

If the government repays its debt, it has access to the international credit market and can borrow at an endogenous bond price of $q(d', z, g)$:

$$V_{pay}(d, z, g) = \max_{c,l,d'} \{u(c, l) + \beta E[V(d', z', g')|z, g]\}$$
$$s.t. \quad c + g + d = e^z f(k, l) + q(d', z, g)d' \tag{2}$$

where $u(\cdot)$ is the representative household's utility function. $c$ and $l$ are consumption and labor respectively. $f(\cdot)$ is the production function of the economy where capital $k$ is kept a constant for computational ease.

If the government defaults, it faces a twofold punishment: temporary exclusion from the international credit market and additional output cost. Bulow and Rogoff [38] point out that if the cost of default is merely no access to international capital markets and if a country can save at the international market interest rate, default happens with probability one and no country should be able to borrow anything. This is the well-known Bulow-Rogoff paradox. Bulow and Rogoff [38] thus propose the idea of additional punishments as a rationale behind why countries repay. With probability $\theta$, the country is able to reenter the market in the next period. When the country reenters, its debt is cleared and is equal to 0. Hence, the value of default can be written as:

$$V_{def}(z, g) = \max_{c,l} \{u(c, l) + \theta \beta E[V(0, z', g')|z, g] + (1 - \theta)\beta E[V_{def}(z', g')|z, g]\}$$
$$s.t. \quad c + g = (1 - \gamma)e^z f(k, l) \tag{3}$$

where $\gamma$ is the proportional output cost.

Foreign lenders are assumed to be risk neutral; they are willing to lend to the sovereign as long as they receive an expected return of the international risk-free rate, $r^*$. Define the default function

$$D(d', z, g) = \begin{cases} 1 & if \ V_{def}(z, g) > V_{pay}(d', z, g) \\ 0 & otherwise \end{cases} . \tag{4}$$

Then the risk neutrality of foreign lenders implies that

$$q(d', z, g) = \frac{E[1 - D(d', z, g)]}{1 + r^*} . \tag{5}$$

Finally, the country's gross interest rate is the inverse of the discount bond price: $\frac{1}{q} = 1 + r^c$, and the sovereign spread is defined as the difference between the country interest rate and the risk-free rate, $r^c - r^*$.

## Equilibrium

Definition: The recursive equilibrium for this economy is defined as a set of functions for decision rules $c(d, z, g)$, $l(d, z, g)$, $d'(d, z, g)$, $D(d', z, g)$, values $V(d, z, g)$, $V_{pay}(d, z, g)$, $V_{def}(z, g)$, and bond price $q(d', z, g)$ such that

1. Taking as given the bond price function $q(d', z, g)$, $c(d, z, g)$, $l(d, z, g)$, $d'(d, z, g)$, $D(d', z, g)$,$V(d, z, g)$, $V_{pay}(d, z, g)$, and $V_{def}(z, g)$ solve the sovereign's optimization problem Eq (1).

2. Bonds prices $q(d'(d, z, g), z, g)$ satisfy foreign lenders' risk neutrality condition Eq (5).

## Calibration and functional forms

The model is calibrated to Argentina's quarterly data over 1993Q1–2012Q4. The risk-free interest rate is set to be equal to the return on U.S. Treasury bonds for a maturity comparable with that of the bonds included in the EMBI Global. Following Bunda, Hamann, and Lall [39], 10-year U.S. Treasury Bond quarterly yield (1.66%) is used as the benchmark risk-free rate. The production function is Cobb–Douglas:

$$f(k, l_t) = k^{\alpha} l_t^{1-\alpha} \tag{6}$$

with a capital share $\alpha$ of 0.36. The constant capital is set to 0.975 so that the quarterly steady state capital-output ratio is 2, which is the converging capital-output ratio for Argentina in Kydland and Zarazaga [40].

The household's period utility function takes the GHH form:

$$u(c_t, l_t) = \frac{\left(c_t - \kappa \frac{1}{\omega} l_t^{\omega}\right)^{1-\sigma}}{1 - \sigma} \tag{7}$$

where the coefficient of relative risk aversion, $\sigma$, is set to 2. GHH preferences are commonly used in small open economy models in order to match the countercyclicality of net exports observed in the data. See Neumeyer and Perri [41]. The curvature parameter of labor supply, $\omega$, is set to 1.455 so that Frisch wage elasticity is equal to 2.2. $\kappa$ is the relative importance parameter of leisure and is calibrated to target a steady state labor supply of 0.33.

A small subjective time discount factor is needed for default to occur in equilibrium. In the literature, $\beta$ ranges from 0.72 to 0.95. Here we set $\beta$ equal to 0.8 as in Aguiar and Gopinath [2]. The probability of reentry $\theta$ = 10% (implying an average stay in autarky of 2.5 years) and the loss of output in autarky $\gamma$ = 2% are also taken from Aguiar and Gopinath [2].

Both the productivity shock and the government spending shock are assumed to follow an AR (1) process:

$$z_t = \mu_z(1 - \rho_z) + \rho_z z_{t-1} + \varepsilon_t^z, \qquad \varepsilon_t^z \sim N(0, \sigma_z^2) \tag{8}$$

$$log(g_t) = \mu_g(1 - \rho_g) + \rho_g log(g_{t-1}) + \varepsilon_t^g, \qquad \varepsilon_t^g \sim N(0, \sigma_g^2) \tag{9}$$

where $\mu_z$ is set to 0; $\rho_z$ and $\sigma_z^2$ are set to match the autocorrelation and standard deviation of Argentine GDP. $\rho_g$ and $\sigma_g^2$ are obtained directly from the AR(1) regression of the (detrended) log of government spending. Finally, $\mu_g$ is calculated to be -2.737 so that the steady state ratio of government spending to output is equal to 13.3%, which is the average ratio of government spending to output over the sample period. Table 1 provides a summary of the parameter values used in the computation.

## Simulation results

This section presents the impulse responses of the model economy following a positive government spending shock.

Fig 1 shows the impulse responses when we feed the path of government spending from the panel VAR into the model. Productivity is kept constant. External debt increases by 0.57% on impact at quarter 0 and stays positive for 5 periods after the shock before reverting back to its steady state. The ratio of current account to GDP falls on impact and then turns positive, which behaves similarly to what is observed from the data of the next section.

The impact response of the spread is 9 basis points. The increase in the sovereign spread can be explained by the increase in external debt, since a higher level of external debt raises the

**Table 1. Parameter values.**

| Parameters | | Value | Target |
|---|---|---|---|
| Risk-free interest rate | $r^*$ | 1.66% | US 10-Year quarterly yield |
| Capital share in output | $\alpha$ | 0.36 | Standard |
| Coefficient of relative risk aversion | $\sigma$ | 2 | Standard |
| Curvature parameter of labor supply | $\omega$ | 1.455 | Frisch wage elasticity (2.2) |
| Relative importance parameter of leisure | $\kappa$ | 1.565 | $l = 0.33$ |
| Subjective time discount factor | $\beta$ | 0.8 | literature: 0.72–0.95 |
| Reentry Probability | $\theta$ | 0.1 | 2.5-year exclusion |
| Output cost if default | $\gamma$ | 0.02 | Standard |
| Serial correlation of $g$ | $\rho_g$ | 0.916 | Regression estimate |
| Variance of innovation to $g$ process | $\sigma_g^2$ | $0.018^2$ | Regression estimate |
| Serial correlation of $z$ | $\rho_z$ | 0.960 | GDP autocorrelation |
| Variance of innovation to $z$ process | $\sigma_z^2$ | $0.005^2$ | GDP standard deviation |

probability of default. However, the responses are much smaller than those in the data. This difference is due to the one-to-one mapping between the default probability and the sovereign spread. Recall that the sovereign spread is the difference between the country's interest rate and the risk-free rate, $\frac{1+r^*}{1-E[D(d',z,g)]} - (1 + r^*)$. A spread of 132 basis points, together with a risk-

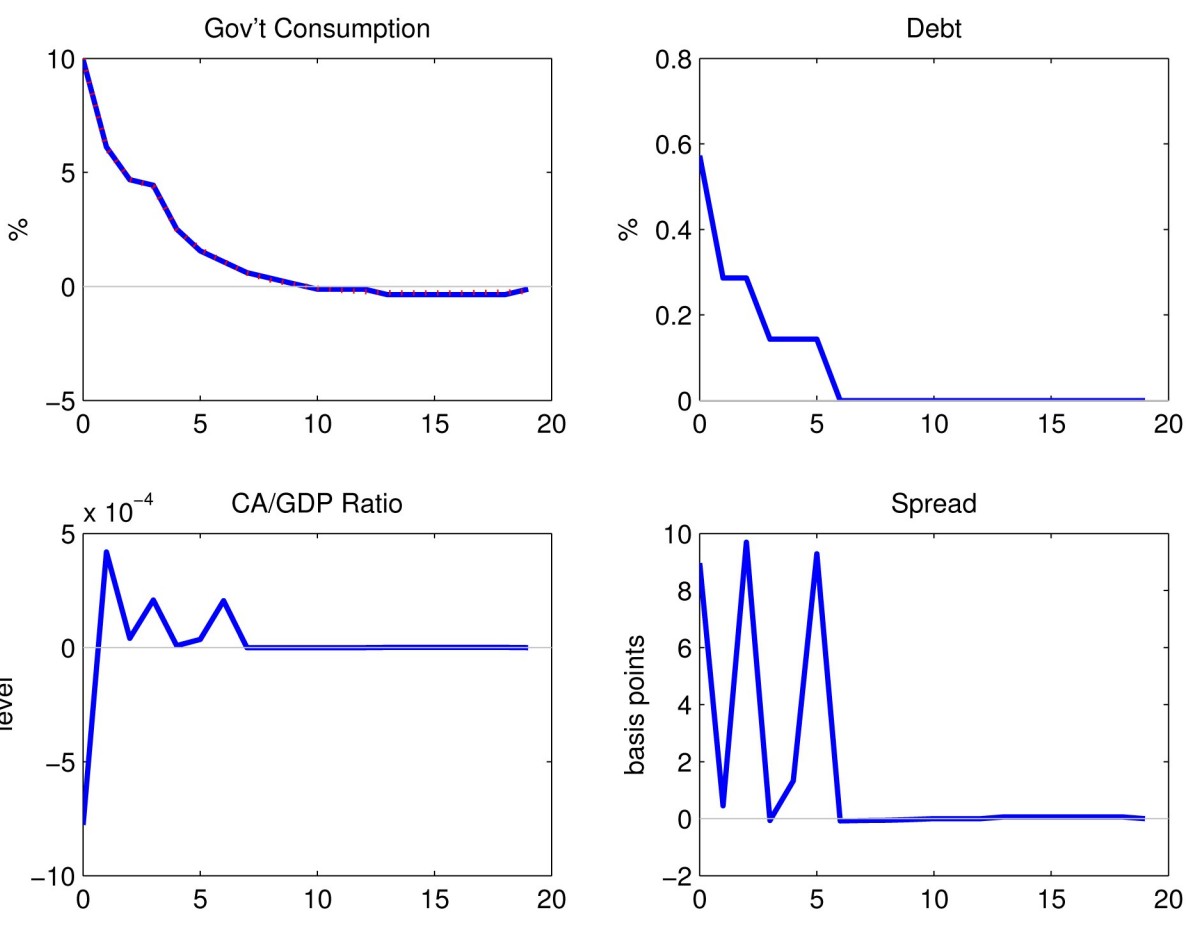

**Fig 1. Impulse responses to the government spending shock.**

free rate of 1.66%, implies that $E[D(d', z, g)] = 0.0128$, which is 1.5 times the default frequency of Argentina (0.008294). Therefore, even if we calibrate $\beta$ to match the default frequency, the model response of the sovereign spread would still be much lower compared to the response of the J.P. Morgan Global spread. The puzzle that a low default probability coexists with a high level of spread in bonds is well known in the finance literature on corporate defaultable bonds. Arellano [3] uses risk averse pricing kernel to achieve a higher level of spread in her model. Solving the puzzle of huge increase in the spread is left for future work.

## Empirical analysis

### Data

This paper uses quarterly-frequency data for 18 developing countries: Argentina, Brazil, Bulgaria, Chile, Colombia, Croatia, Ecuador, El Salvador, Hungary, Lithuania, Malaysia, Mexico, Peru, Poland, South Africa, Thailand, Turkey, and Uruguay. The J.P. Morgan Emerging Markets Bond Index Global ("EMBI Global") spread has been used to evaluate the sovereign default risk of a developing country. EMBI Global tracks total returns for U.S. dollar-denominated Brady bonds, Eurobonds, and traded loans issued by emerging market sovereign and quasi-sovereign entities. Spread over Treasury is calculated as the difference between the yield to maturity of each bond (i.e. the internal rate of return of the bond instrument) and the yield to maturity of the corresponding point on the U.S. Treasury spot curve. For more details, see Bunda, Hamann, and Lall [39] and Jaramillo and Tejada [42]. The choice of countries is based on the availability of J.P. Morgan's EMBI Global spread data and the individual country's quarterly government spending data. Except for Lithuania and Malaysia, all other countries have experienced sovereign debt default or restructuring as documented in Table 2 in the S1 Appendix. On average, a typical emerging economy defaulted 5 times during the last 211 years, indicating an annual default frequency of 0.02396, or 0.00599 in quarterly terms.

Quarterly external debt data are taken from the Joint External Debt Hub. Series "liabilities to BIS banks, locational, total" is used as external debt in the regression because it is the longest data series available, dating back to 1993Q3. The paper's results are robust to choosing other external debt series covering a shorter time span. Those series include "liabilities to BIS banks, consolidated, total", "cross-border loans from BIS reporting banks", "international debt securities, all maturities", and "multilateral loans, total". All the other variables (GDP, government consumption, private consumption, the current account, and the real effective exchange rate) come from the same data source as in Ilzetzki, Mendoza, and Végh [6].

### Estimation methodology

The following system is estimated by Panel OLS regression with fixed effects:

$$AY_{n,t} = \sum_{k=1}^{K} C_k Y_{n,t-k} + Bu_{n,t}, \tag{10}$$

where $n$, $t$, and $k$ represent a country index, a time index and lag length, respectively.

$Y_{n,t} = (gc_{n,t}, gdp_{n,t}, prcon_{n,t}, debt_{n,t}, CA_{n,t}, spread_{n,t}, REER_{n,t})$, where $gc_{n,t}$ is real government consumption, $gdp_{n,t}$ real GDP, $prcon_{n,t}$ real private consumption, $debt_{n,t}$ external debt, $CA_{n,t}$ the ratio of the current account balance to GDP, $spread_{n,t}$ the EMBI Global spread, and $REER_{n,t}$ the real effective exchange rate. Government and private consumption, GDP, external debt, and REER are in natural logarithms. Variables that show seasonal patterns are deseasonalized using the SEATS algorithm. The non-stationary government and private consumption, GDP, and debt data are included in the regression as deviations from their

quadratic trend. The current account ratio is included in levels and the spread is in basis points. REER is included in first differences. $u_{n,t}$ is a vector of orthogonal, i.i.d. structural shocks with $Eu_{n,t} = 0$ and $Eu_{n,t}u'_{n,t} = I$. The coefficient matrices $A$, $B$ and $C_k$ are assumed to be invariant across time and countries.

Additional restrictions must be imposed to estimate the coefficients. Government spending is ordered first. This is based on the assumption that it takes more than a quarter for fiscal authorities to respond to shocks to other macroeconomic variables. We use Cholesky decomposition for the other restrictions required to estimate the matrices. Since only government spending shocks are of interest in this paper, the ordering of the remaining variables is irrelevant.

The baseline regression includes data for all the 18 developing countries from 1993Q1 to 2012Q4. The dataset is then divided into fixed exchange rate periods versus flexible exchange rate periods, as well as Latin American countries versus European countries. The sample size of panel data for Asian countries is relatively small, so the confidence bands are quite wide for Asian countries. A lag length of four is chosen for all regressions so that the difference between regressions should not result from a different lag length. The results are robust to choosing lag length according to lag order selection criteria: Akaike Information Criterion, Schwarz Criterion, Hannan-Quinn Criterion, Likelihood Ratio Test and Wald Test. Actually this paper's results are robust to choosing any alternative lag length from 1 to 8.

## All sample countries

Fig 2 shows the impulse responses to a 10% government spending shock at quarter 0. The responses of private consumption are not shown because the patterns are very similar to those of GDP. One period refers to one quarter. Shaded areas represent 68% confidence intervals. GDP increases by 1.33% on impact and remains positive till quarter 8. However, the responses of GDP are negative in the long run, implying that the increase in government spending is more-than-fully crowded out by the decrease in the other components of GDP.

The real effective exchange rate (REER) appreciates on impact. In the short run, an increase in government spending raises the demand for domestic goods, leading to the appreciation of domestic goods. Starting from quarter 7, the responses of REER remain significantly below zero. Thus the long-run effect of government spending shocks is a depreciation of domestic goods. Correspondingly, the current account balance to GDP ratio first falls but then turns positive after quarter 10. The reversal in the current account and the REER could be attributed to the deterioration of the domestic country's debt position. Foreign agents may become reluctant to lend to the domestic country because of a higher default risk. In extreme cases, there might even be sudden stops, i.e. a sudden slowdown in capital inflows into the domestic country.

External debt increases by 1.19% on impact. On average, external debt increases by 3.45% in the year following the government spending shock. At quarter 20, the response of external debt turns negative. There are two possible reasons for this change of sign in external debt. First, governments may have engaged in fiscal consolidation efforts as reflected by the negative responses of government consumption since quarter 13. Second, governments may intend to reduce their borrowing because of higher cost, as shown by a higher spread.

With a delay, the EMBI Global spread rises significantly and remains positive throughout the simulation. Within two years following the shock, the spread increases by an average of 25 basis points. The response peaks at 132 basis points 14 quarters after the spending shock. Overall, the expansionary spending of a sovereign is accompanied by an increased level of sovereign spread hence default risk.

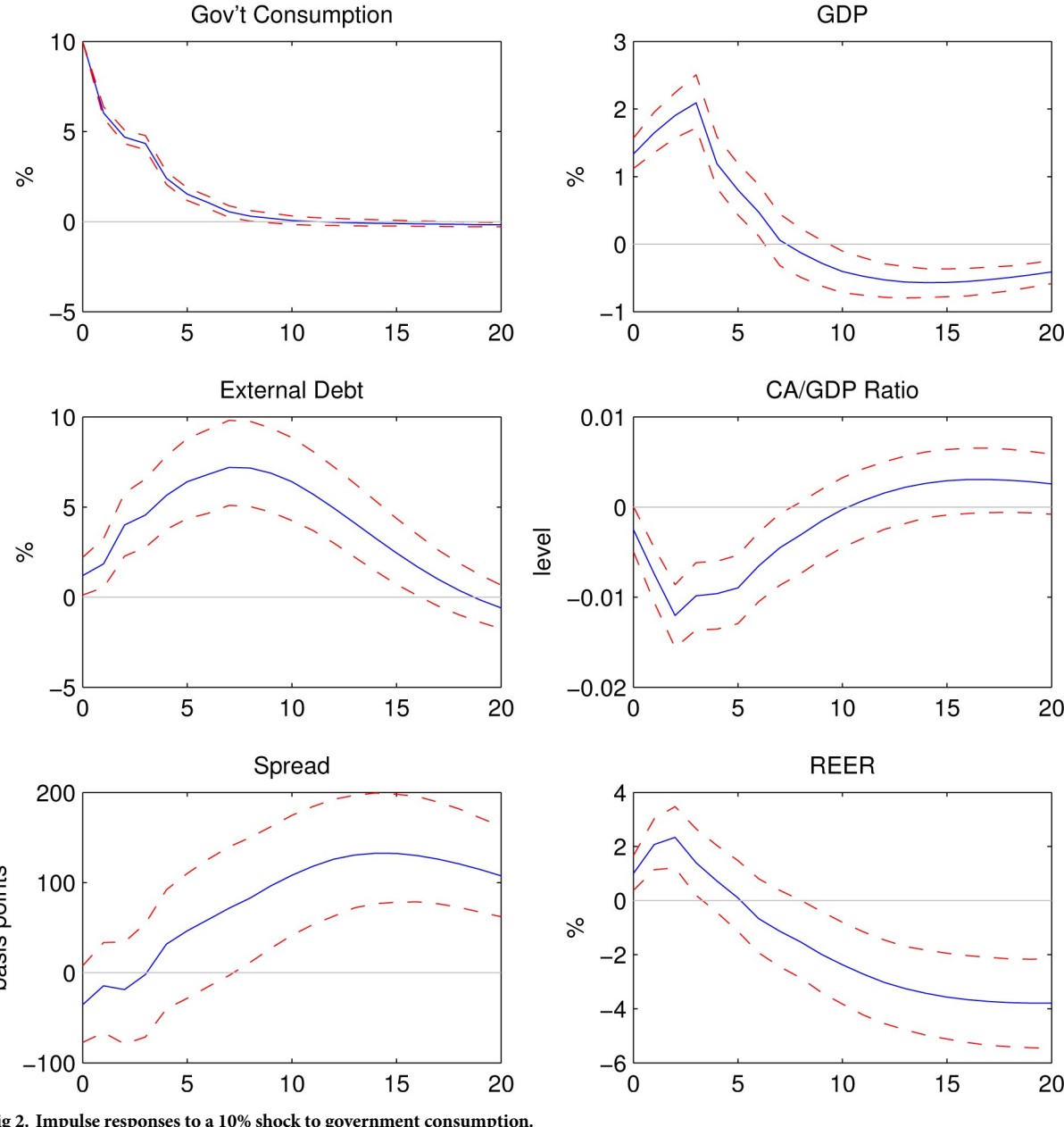

**Fig 2. Impulse responses to a 10% shock to government consumption.**

### Fixed and flexible exchange rate regimes

In this section each of the 18 developing countries is divided into episodes of fixed exchange rate regime and episodes of flexible exchange rate regime. The de facto classification of Ilzetzki, Reinhart, and Rogoff [43] is used to determine the exchange rate regime of each country in each quarter. Fixed exchange rate regime in a given country includes the episodes with no legal tender, hard pegs, crawling pegs, and de facto or pre-announced bands or crawling bands with margins no larger than ±2%. All other episodes in this country are classified as flexible. See the S1 Appendix for a detailed list of episodes of de facto fixed and flexible exchange rates in each country. The results are presented in Fig 3. Not surprisingly, for the fixed exchange

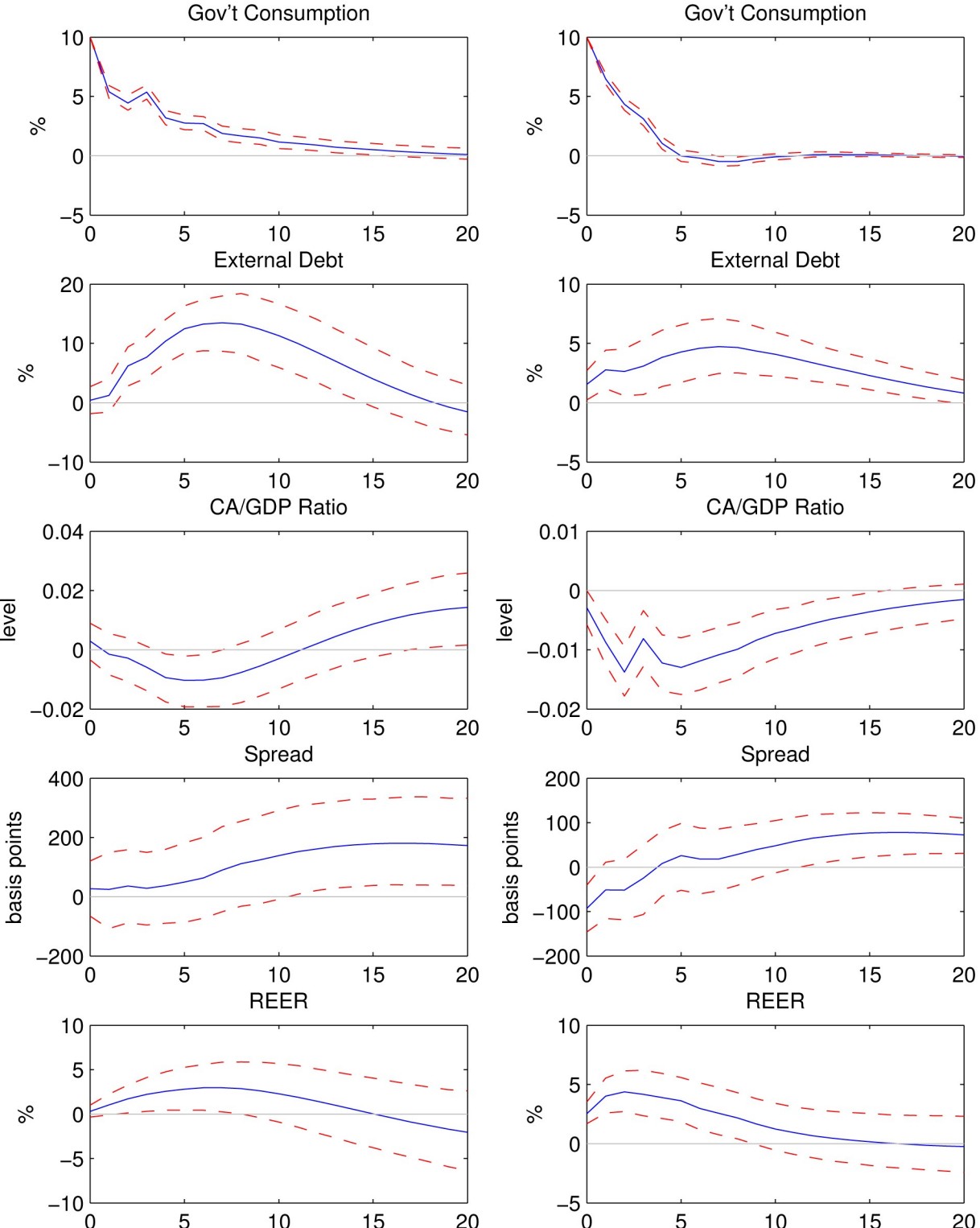

**Fig 3. Impulse responses to a 10% shock to government consumption in episodes of fixed exchange rates (left) and flexible exchange rates (right).**

rate sub-group, the impact response of REER is insignificant. In contrast, there is a significant appreciation of REER on impact for the flexible exchange rate counterpart. The responses of the current account balance to GDP ratio are also quite different for these two groups: no significant changes in the fixed exchange rate regime, where the confidence bands contain zero in most of the simulation periods, versus significant worsening of this ratio in the flexible regime. Moreover, the impact response of GDP in the flexible exchange rate regime group is larger (1.76%) than that in the fixed regime group (1.14%).

It is interesting to notice that the magnitude of spread increase in the fixed exchange rate regime is larger than that in the flexible regime. This suggests that when an expansionary government spending shock hits the economy, the increase of default risk in countries with flexible exchange rate regimes tends to be smaller compared to those with fixed exchange rate. Hence the latter countries face a larger increase in the interest rate at which they borrow in the international credit market.

### Latin American and European countries

Fig 4 compares the impulse responses to a 10% shock to government spending in Latin American countries and European countries in the sample. In Latin American countries, government spending shocks are more persistent. With a delay, the EMBI Global spread shows significant increase in both groups. However, the increase is much larger for Latin American countries (peaking at 248 basis points) than for European countries (peaking at 90 basis points), implying that default risk rises by a much larger margin in Latin American countries after a fiscal expansion.

In Latin American countries, the expansion of government spending leads to exchange rate appreciation throughout the simulation. The response of the ratio of the current account to GDP is negative in the long run, implying that current account balance worsens in the long run. In European countries, there is a significant depreciation of REER in the long run and the current account improves in the long run. The stimulative effect on GDP is larger in European countries than in Latin American countries: GDP increases by 2.20% on impact in the former and by 1.41% in the latter.

## Conclusion

In this paper, we have built a general equilibrium small open economy model that features a higher sovereign default risk in response to positive government spending shocks. VAR evidence from a panel of 18 emerging countries shows that the sovereign spread increases following a positive government spending shock, confirming the theoretical predictions from the general equilibrium model. The J.P. Morgan Global spread rises by an average of 25 basis points within two years following a 10% spending shock and the peak value is as high as 132 basis points. In the meantime, there is a significant increase (3.45% on average) in external debt in the year following the shock. The short-run and long-run responses are different for the other four macroeconomic variables under consideration. In the short run, the responses of GDP and private consumption are positive; real effective exchange rate appreciates and the current account deteriorates. In the long-run, those responses change sign.

Fiscal polices in emerging economies tend to be procyclical [13, 44]. Developing countries tend to spend more in good times and less in bad times. The findings of this paper have important implications for policymakers. A fiscal stimulus temporarily boosts GDP. However, as a nation's sovereign debt piles up, it has to pay a higher interest rate when borrowing from abroad, since the default risk is higher. This could exert a negative impact on the nation's economic activities in the long run. This external debt channel weakens the usefulness of

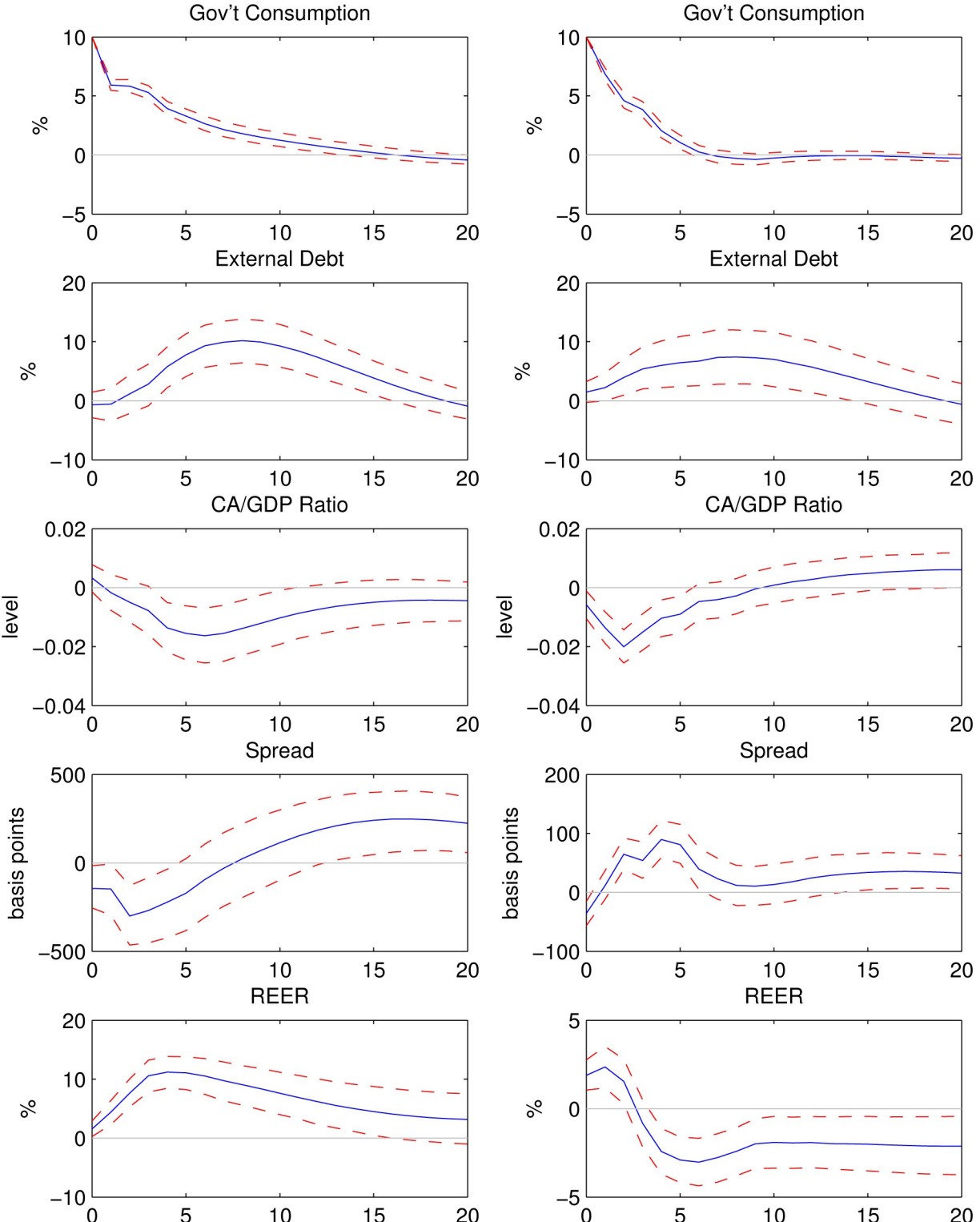

**Fig 4. Impulse responses to a 10% shock to government consumption in Latin American countries (left) and European countries (right).**

government spending. If developing countries could instead save in good times, then it would attenuate the negative impact of the external debt channel. In this way, fiscal polices could be more effective.

The current paper does not consider the financing method of government spending in developing countries. It is suggested for future researchers to distinguish among tax financed spending, debt financed spending, and money financed spending. Moreover, since government spending multiplier depends on the degree of economic development [12], it will be useful for future researchers to distinguish among low-income economies, lower-middle-income economies, and upper-middle-income economies, as data become available. Finally, it would be important to explore the interplay between monetary and fiscal policies in a sovereign default model. This direction is also left for future research.

## Supporting information

**S1 Appendix. Computation and data.**
(PDF)

**S1 Dataset. Dataset.**
(XLSX)

## Acknowledgments

We would like to thank Bill Dupor, Paul Evans, Pok-Sang Lam, and Byoung Hoon Seok for valuable comments and suggestions.

## Author Contributions

**Methodology:** Jingchao Li.

**Software:** Ming Jiang.

**Writing – original draft:** Jingchao Li.

**Writing – review & editing:** Jingchao Li.

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
