## [Decision Letter · Decision Letter 0]

31 May 2023

PONE-D-23-08520Government Spending Shocks and Default Risk in Emerging MarketsPLOS ONE

Dear Dr.  Li,

Thank you for submitting your manuscript to PLOS ONE. After careful consideration, we feel that it has merit but does not fully meet PLOS ONE’s publication criteria as it currently stands. Therefore, we invite you to submit a revised version of the manuscript that addresses the points raised during the review process.

We look forward to receiving your revised manuscript.

Kind regards,

Ayesha Afzal, PhD

Academic Editor

PLOS ONE

Journal Requirements:

"Jiang thanks the National Natural Science Foundation of China (Grant 72103135 and Grant 72231003) for research support. Li thanks the National Social Science Fund of China (Grant 21AZD036) and the National Natural Science Foundation of China (Grant 72274062) for research support."

Reviewers' comments:

Reviewer's Responses to Questions

**Comments to the Author**

1. Is the manuscript technically sound, and do the data support the conclusions?

Reviewer #1: Yes

Reviewer #2: Yes

2. Has the statistical analysis been performed appropriately and rigorously? 

Reviewer #1: I Don't Know

Reviewer #2: Yes

3. Have the authors made all data underlying the findings in their manuscript fully available?

Reviewer #1: Yes

Reviewer #2: No

4. Is the manuscript presented in an intelligible fashion and written in standard English?

Reviewer #1: Yes

Reviewer #2: No

5. Review Comments to the Author

Reviewer #1: This article is about the role of government spending in default risk in emerging economies. A gap has been underlined to justify the study but this contribution could have been written in a better way. Relevant literature has been incorporated but there is no recent article. The literature is not explained well so it is hard for a reader to understand the importance of the studied variables and relationships between them. The results are provided but without explanation or discussion. No implications have been given to show how these findings can be utilised by economic and public policy makers, teachers and researchers. The conclusion is weak.

Reviewer #2: 1) Abstract

a) Abstract should be more concrete and therefore needs realignment.

a) The keywords are more or less repetition of the research title. It is suggested to revisit the keywords.

2) Introduction:

a) The paragraph 1 lacks references. All the statements, facts and rationale put forth should be supported with proper in text citations.

b) The contribution of the research is missing. It is recommended to add the contributions and justify your contribution. For example, review the following papers and gain insight:

i) Cai, L., Firdousi, S. F., Li, C., & Luo, Y. (2021). Inward foreign direct investment, outward foreign direct investment, and carbon dioxide emission intensity-threshold regression analysis based on interprovincial panel data. Environmental Science and Pollution Research, 1-14.

ii) Li, C., Firdousi, S. F., & Afzal, A. (2022). China’s Jinshan Yinshan sustainability evolutionary game equilibrium research under government and enterprises resource constraint dilemma. Environmental Science and Pollution Research, 29(27), 41012-41036.

iii) Mirza, N., Umar, M., Afzal, A., & Firdousi, S. F. (2023). The role of fintech in promoting green finance, and profitability: Evidence from the banking sector in the euro zone. Economic Analysis and Policy, 78, 33-40.

iv) Awais, M., Afzal, A., Firdousi, S., & Hasnaoui, A. (2023). Is fintech the new path to sustainable resource utilisation and economic development? Resources Policy, 81, 103309.

3) Theoretical background and hypothesis development

The introduction and literature review section are merged as one. It is fine although recommend to divide it under two parts. Moreover, there is a need for more recent studies ranging from 2018-2023 to support the study hypothesis properly. The entire study is too scanty and the related literature is not exhausted.

a) The first part of introduction is supported with many facts and figures but yet again there is quite old reference to support the statements.

b) The literature review is also weak. The hypothesis statements should be added and explained well. The theoretical connections should be established to justify the research variables and proposed hypothesis. Moreover, update references are required.

4) Data and methodology

The econometric modelling presented is good. However, it is suggested proper reasoning for opting VAR should be provided with references.

5) Results and Discussion

a) There is hardly any referencing to support the results. The results are although quite significant and interesting. However, there needs to be proper discussion to support/reject the hypothesis made earlier.

6) Conclusion and Implications

a) I would highly recommend inclusion of this section in the revised manuscript. You can review the following paper for this section:

i) Li, C., Firdousi, S. F., & Afzal, A. (2022). China’s Jinshan Yinshan sustainability evolutionary game equilibrium research under government and enterprises resource constraint dilemma. Environmental Science and Pollution Research, 29(27), 41012-41036.

ii) Afzal, A., Firdousi, S. F., Waqar, A., & Awais, M. (2022). The Influence of Internet Penetration on Poverty and Income Inequality. Sage Open, 12(3), 21582440221116104.

b) It is suggested to add one paragraph on study limitations and future research direction.

Last but not the least, there are many grammatical errors in the entire manuscript. It is highly recommended to review the document in detail. It is recommended to use professional reference and citation software so that paper seems well presented.

6. PLOS authors have the option to publish the peer review history of their article (what does this mean?). If published, this will include your full peer review and any attached files.

Reviewer #1: No

Reviewer #2: **Yes: **Dr Saba Fazal Firdousi

---

## [Editor Report · Decision Letter 1]

4 Jul 2023

Government Spending Shocks and Default Risk in Emerging Markets

PONE-D-23-08520R1

Dear Dr. Li,

We’re pleased to inform you that your manuscript has been judged scientifically suitable for publication and will be formally accepted for publication once it meets all outstanding technical requirements.

Kind regards,

Ayesha Afzal, PhD

Academic Editor

PLOS ONE

Additional Editor Comments (optional):

All reviewers comments have been addressed appropriately.
---

## [Editor Report · Acceptance letter]

10 Jul 2023

PONE-D-23-08520R1 

Government Spending Shocks and Default Risk in Emerging Markets  

Dear Dr. Li:

I'm pleased to inform you that your manuscript has been deemed suitable for publication in PLOS ONE. Congratulations! Your manuscript is now with our production department. 

Kind regards, 

on behalf of

Dr. Ayesha Afzal 

Academic Editor

PLOS ONE